# Weakly-Supervised Trajectory Segmentation for Learning Reusable Skills

## Abstract

Learning useful and reusable skill, or sub-task primitives, is a long-standing problem in sensorimotor control. This is challenging because it's hard to define what constitutes a useful skill. Instead of direct manual supervision which is tedious and prone to bias, in this work, our goal is to extract reusable skills from a collection of human demonstrations collected directly for several end-tasks. We propose a weakly-supervised approach for trajectory segmentation following the classic work on multiple instance learning. Our approach is end-to-end trainable, works directly from high-dimensional input (e.g., images) and only requires the knowledge of what skill primitives are present at training, without any need of segmentation or ordering of primitives. We evaluate our approach via rigorous experimentation across four environments ranging from simulation to real world robots, procedurally generated to human collected demonstrations and discrete to continuous action space. Finally, we leverage the generated skill segmentation to demonstrate preliminary evidence of zero-shot transfer to new combinations of skills. Result videos at https://sites.google.com/view/trajectory-segmentation/.

## 1 Introduction

Humans have an uncanny ability to generalize from one task to another using either few, or at times, no new examples. This wouldn't be possible if they were to learn each new task from scratch. Humans rather extract reusable *skills* from already learned tasks and compose them to generalize to new tasks seamlessly. However, learning such repeatable skills has been a long standing challenge in sensorimotor control, partly because it is hard to define what constitutes a useful skill in itself.

One way to layout the scope of a skill is either by designing a corresponding reward function, or collecting expert demonstrations. For instance, consider a skill of reaching for an object. One can easily learn a policy for this skill by either using reinforcement learning with l2 distance as reward (Sutton & Barto, 1998), or by imitation learning from kinesthetic demonstrations (Argall et al., 2009; Hussein et al., 2017). However, neither of these approaches provide a natural form of supervision because the way this skill is performed in isolation can be drastically different from the way it could be used as part of some end task. For instance, reaching for a cup for pushing is very different from the way one would reach for a cup to pick it up for pouring. Therefore, defining a skill directly in a supervised manner could easily lead to biased set of examples.

A promising alternative is to learn skills that are already embedded in some useful end tasks. Previous works have explored this in the context of an agent's own exploration (Eysenbach et al., 2018; Nair et al., 2018; Pathak et al., 2018), where, the agent learns goal conditioned skill policies using data collected during its exploration phase. These skills are then used to plan for novel tasks at inference. However, exploration in itself is an open research problem, and hence, such approaches have difficulty in scaling to complex skills.

In this work, we follow an alternative paradigm where we extract skills from a collection of human demonstrations gathered to perform different end tasks. A straightforward way to extract reusable skills would be to get an expert to label each time-step of demonstrations with the corresponding skill label. However, this per-step segmentation supervision is tedious for the expert and too expensive to scale. Moreover, such a labeling would be biased towards what expert thinks is the right segmentation than towards a segmentation which helps learn the skills better. This leads to the question: is it

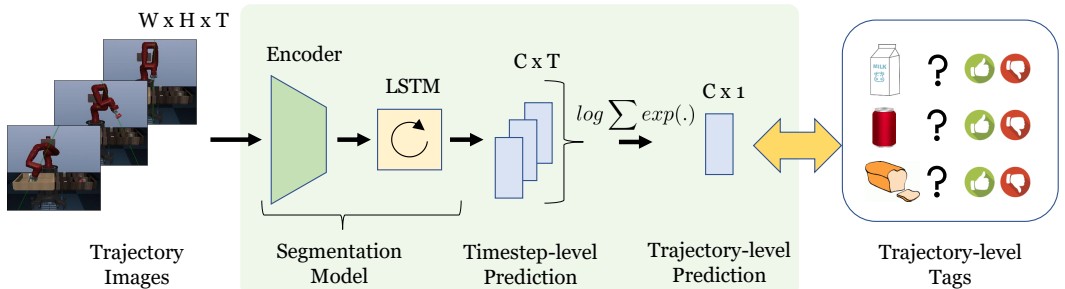

Figure 1: We propose a weakly-supervised approach for segmenting demonstrations into skill primitives, aka, sub-tasks. Our approach is end-to-end trainable, works directly from raw sensory data (e.g., images) and only requires the knowledge of class of primitive sub-tasks performed during the demonstration to accomplish the end task, without any need of segmentation or ordering of primitives. The key idea is to make per-time step prediction corresponding to each sub-task and accumulate them to generate a trajectory-level predictions which are then trained to match trajectory-level tags. Our segmentation model consists of an encoder followed by a recurrent LSTM model to capture time-series dependency. We use `log-sum-exp` to perform smooth accumulation of time-step predictions into trajectory-level predictions.

possible to use the expert knowledge to only know the types of skills present in demonstration and figure out the segmentation from the data itself?

Inspired by the classic work in multiple instance learning (MIL) (Andrews et al., 2002), we propose a weakly-supervised approach for segmentation of human demonstration trajectories into primitive repeatable skills. Our approach assumes the access to only trajectory-level labels, i.e., what primitive skills, aka sub-tasks, are present in the demonstration. The key insight is to learn a primitive skills classifier model conditioned on the input sensory data (e.g. demonstration images), and incorporate a per-time step reasoning structure in this classifier. An overview of our approach is shown in Figure 1. Our model generates per time-step primitive skill label prediction estimates which are then accumulated via differentiable function to generate trajectory-level predictions. In contrast to classic MIL, where only the most confident prediction across time-steps is trained, our full model is trained end-to-end using trajectory-level multi-class loss directly from raw sensory images.

However, we are training with only trajectory-level supervision, then why should our per-step predictions of segmentation model converge to meaningful skill primitives? Data comes to rescue! Since our model trains across a variety of demonstrations, and hence, it would have seen plenty of demonstration trajectories that contain a certain skill primitive (positives) and plenty that do not (negatives). The classification loss would force the segmentation model to focus on discriminative cues that are common across all positives, and absent from negatives. These discriminative cues corresponding to each skill would encourage the per time-step predictions to gradually correspond to the correct ground truth time-step labels.

We evaluate our approach in four different environments: (a) As proof of concept, we start with a simple 2D navigation task in grid-world setup where the demonstrations are programatically generated. (b) We then discuss result in robotics setup with continuous control actions, in particular, Jaco robotic arm performing button pushing demonstrations in a touch pad with procedurally generated demonstrations. (c) We test our approach in a robotic setup with actual human demonstrations collected on the RoboSuite benchmark. (d) Finally, we evaluate on a real robot dataset with actual human demonstrations collected kinesthetically. Across all these environments, our approach outperforms the other variants of MIL and achieves decent segmentation accuracy. We then show zero-shot generalization to tasks containing novel permutations of skills in Jaco environment.

## 2 METHOD: SEGMENTING DEMONSTRATIONS INTO SKILL PRIMITIVES

Given a collection of human demonstration trajectories, our goal is to learn a labeling for skill primitives at each time-step of the sequence, i.e., per time-step skill segmentation. Let $X$ be a human demonstration trajectory denoted by $X = \{x_1, x_2, x_3 \ldots x_T\}$, where $T$ is the length of the demonstration and $x_t$ denotes a tuple of observation (which is raw sensory image in our case) at time $t$

and action taken from it. Note that the action data is optional for the purpose of skill segmentation, but can be useful post segmentation for learning skill policies via imitation. Let $Y = \{y_1, y_2, y_3 \ldots y_T\}$ be the latent ground truth labeling of skill primitives in the sequence. Each label $y_t$ belongs to one of the $k$ labels from a set of all skill classes $\mathcal{C} = \{1, \ldots, k\}$, i.e., $y_t \in \mathcal{C}$. These per time-step labels are not only tedious for expert to annotate, but also difficult to scale. In this work, we do not assume access to $y_t$ during training, and learn the per time-step segmentation in a weakly-supervised fashion by only using trajectory-level 1-bit label during training, i.e., whether a skill class is present in the trajectory or not. After training, our model is able to directly segment demonstrations at inference, without requiring any labels of any kind. An overview of our method is shown in Figure 1.

The marginal probability of a skill primitive at each time-step of demonstration can be written as $P(y_t|\theta, \{x_t, x_{t-1} \ldots, x_1\})$ where $\theta$ is the parameter vector of the segmentation model represented by a neural network in our case. If we had access to the true sequence labels $Y$, the network parameters $\theta$ can be easily learned via maximum log-likelihood by representing the probability as follows:

$$P(y_t|\theta, \{x_t, x_{t-1} \ldots, x_1\}) = \frac{1}{Z_t} \exp\left(f(y_t; \theta, \{x_t, x_{t-1} \ldots, x_1\})\right) \tag{1}$$

where $Z_t$ is the partition function at $t$, defined as $Z_t = \sum_{k \in \mathcal{C}} \exp\left(f_t(k; \theta, \{x_t, x_{t-1} \ldots, x_1\})\right)$. The output of the function $f_t$ corresponds to the logit score value generated by the neural network. In order to model temporal dependency on across observation time-steps $x_t$, we represent $f(.)$ via a recurrent neural network, in particular, LSTM (Hochreiter & Schmidhuber, 1997).

## 2.1 WEAKLY-SUPERVISED TRAJECTORY SEGMENTATION

We are given a dataset of demonstration trajectories during training, $\mathcal{D} = \{X^1, \ldots, X^n\}$, where $n$ is total number of demonstrations available for training. Each demonstration trajectory is weakly labelled with what skill primitives are contained within the trajectory. Neither do we have access to which time-step densely correspond to which skill primitive, nor to the permutation in which the skills are executed in. Instead, we are only given a set of skill primitive labels $C_X \in \mathcal{C}$ present in the demonstration trajectory $X$.

Although our supervision is only at trajectory-level, we do not directly predict output labeling $C_X$ from input demonstration $X$. Instead, we instill the structure of per-step prediction in our weakly supervised segmentation model by first computing the per-step classification score $f(y_t; \theta, \{x_t, x_{t-1} \ldots, x_1\})$ and then accumulate it across all time-steps to compute the probability of a class in the whole trajectory. This weakly-supervised setup is captured by classical paradigm of multiple instance learning (MIL) (Andrews et al., 2002). At inference, we use this per time-step score to compute the probability of skill primitives at each time $t$ as described in Equation (1). There are multiple ways one could accumulate these per time-step scores discussed as follows.

## 2.2 ACCUMULATION OF TIME-STEP PREDICTIONS

Intuitively, we would like an estimator that could generate an aggregated score for the class depending on how much each time-step votes for that class. We would ideally like the highest score value across time-steps to contribute most to the decision whether a class label $\hat{Y}_X$ is present in the trajectory $X$ or not. One simple way to achieve that is to employ element-wise $\max$ operator, which is also the de facto approach in MIL to accumulate element-level scores. However, this would amount to passing gradients only to the most confident time-step and will completely eradicate the role of other time-steps. This is especially problematic in case of sequential trajectories because no skill primitive will be of only 1 time-step long. Hence, instead of $\max$, we use a soft approximation to it which can take into account the contribution of all time-steps. In particular, we use `log-sum-exp` operator. Given the the logit score $f(y_t; \theta, \{x_t, x_{t-1} \ldots, x_1\})$ at each time-step, the trajectory-level logit score $g$ for class $c \in \mathcal{C}$ is computed as follows:

$$g(c; \theta, X) = \log\left(\sum_{t=1}^{T} \exp\left(f(y_t = c; \theta, \{x_t, x_{t-1} \ldots, x_1\})\right)\right) \tag{2}$$

We perform this operation for all $c \in \mathcal{C}$ and use softmax over $g(c; \theta, X)$ to compute trajectory-level probability distribution $Q(c|X, \theta)$. Finally, the parameters $\theta$ are optimized to maximize $Q(c|X, \theta)$

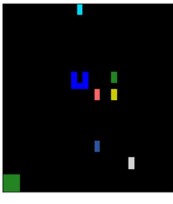 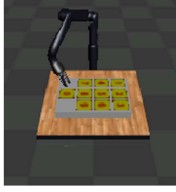 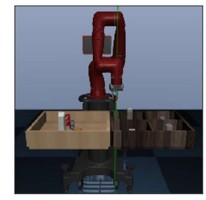 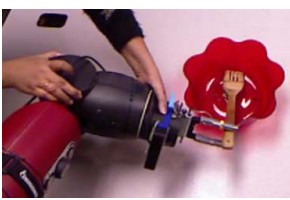

(a) 2D Navigation   (b) Dial Env   (c) RoboSuite Env   (d) MIME Env

Figure 2: We evaluate across four environments with different properties: (a) Discrete 2D Navigation as proof of concept. (b) Dial continuous control: Jaco robotic arm based manipulation with procedurally generated demonstrations. (c) RoboSuite environment: Sawyer robot in simulation with human-collected demonstrations. (d) MIME environment: Baxter robot in real world with human demonstrations. Result videos at https://sites.google.com/view/trajectory-segmentation/.

for each class $c$ with respect to the ground truth trajectory level tags $C_X$. Note that this optimization is fully differentiable through Equation 2, and hence can be optimized via stochastic gradient descent in the end-to-end fashion. Pinheiro & Collobert (2015) has also showed the effectiveness of a temperature-based variant of log-sum-exp operation for semantic segmentation in images.

However, these per-step scores $f$ would be almost uniformly random in the beginning of the training process due to the absence of per-step supervision. Since we are training with only trajectory-level supervision, why should these per-step predictions should ever converge to meaningful skill segmentation? It turns out to be the case because we are learning across large variety of demonstration examples. Hence, for each skill primitive we would have seen plenty of positive as well as negative trajectories. The loss suppresses the negative classes and encourages the positive ones, hence our segmentation model would be forced to focus on discriminative cues that are exclusively common among the trajectories containing the skill primitives and not common to the cues that help distinguish other skills. Since our trajectory-level segmentation is based on a deterministic transformation of per-step predictions, each per-step score will then be forced to focus on those discriminative cues. The discriminative nature encourages the per time-step predictions to slowly drift towards the true latent ground truth segmentation which are not available directly.

## 3   IMPLEMENTATION DETAILS, SETUP AND BASELINES

**Training and Evaluation Details**   Our segmentation model consists of a convolutional neural network encoder of each image of a trajectory and a one layer fully connected encoder of the action to a 32 dimensional feature that is concatenated to the image feature before being fed into an LSTM with 100 hidden units. We train with a batch size of 64 for the Dial, RoboSuite, and MIME environments and a batch size of 128 for the Grid-world environment. All models were trained with Adam with learning rate of 1e-4. For training, we use 50000, 2000, 1000, and 1600 trajectories for 2D Navigation, Dial, RoboSuite, and MIME Environments respectively. We evaluate our method on the training set, a validation set that consists of the same number of skill primitives (sub-tasks) per trajectory as in training, and a test set that consists of more skill primitives per trajectory than seen in training. The segmentation quality is measured by classification accuracy of the learned model averaged across all time-step. The time-step ground truth is only used for evaluation and not for training.

**Baselines**   We compare our approach to different formulations of weakly-supervised classification proposed earlier. None of these methods have been applied to a temporal trajectory segmentation before. In our work, we adapt them for sub-task segmentation. In particular, we compare to: (a) MIL (Andrews et al., 2002): In this baseline, we train the deep segmentation model using Classic MIL formulation as proposed by Andrews et al. (2002). We penalize the most confident time-step in the output corresponding to each sub-class to correctly predict the trajectory-level classes.
(b) FCN-MIL (Pathak et al., 2015b): This approach is an adaptation of MIL for deep-networks. The MIL objective is treated as the loss function for training the segmentation network, but instead of training the most confident output, we train a neighborhood of k-elements near the most confident time-step, where k is a hyper-parameter chosen on the validation set. We found k=3, i.e., one extra time-step on each side of argmax, to work the best across all datasets.
(c) CCNN (Pathak et al., 2015a): This is an alternative approach to tackle the weakly-supervised

setup. Instead of training the most confident output, a pseudo ground-truth per time-step truth is generated in a way that ensures that it is the closest pseudo ground-truth possible which contains only the classes as specified by the trajectory-level tags. One could additionally put threshold constraints on the lower-bound of time-steps devoted to the present classes in the pseudo ground truth. The segmentation is then pushed to this per time-step pseudo labels, and a new pseudo ground-truth is generated after each iteration. This approach provides per time-step supervision using only trajectory level-tags. The CCNN approach was originally applied for per-pixel segmentation of images in computer vision literature (Pathak et al., 2015a), and we adapt it for temporal trajectories.

(d) Random: For sanity check, we randomly pick a sub-task class at every time-step with uniform probability.

(e) Random-Cls: This is a random baseline with privileged class information even at test time. In this case, we only sample random uniformly from the set of classes that are already present in the trajectory. This is not a fair comparison but provides an estimate for the difficulty of the problem.

Note that we use the same network architecture and training details across all baseline implementations. However, we tune each baseline separately in a thorough manner on validation sets. This is crucial to ensure an "apples-to-apples" comparison.

## 4    RESULTS

We evaluate our approach and other baselines on four different datasets with very diverse characteristics. The 2D Grid-world environment has a discrete action space, while the Dial, RoboSuite, and MIME environments have a continuous action space. The Grid-world and Dial environments have demonstrations collected procedurally by hand-designed controllers, while the RoboSuite and MIME environments have demonstrations collected by humans. Human demonstration in RoboSuite are collected via teleoperation and kinesthetically in MIME. Grid world, Dial and Robosuite are in simulation while MIME is from real robot. A snapshot of these environments is shown in Figure 2.

### 4.1    PROOF OF CONCEPT: 2D NAVIGATION IN DISCRETE TOY GRID-WORLD

In the grid-world environment, the action space consists of moving up, moving down, moving left, moving right, and picking up object it is hovering over. There are 5 different types of objects uniquely identified by their color and an end task would consist of picking up some subset of all the objects in a particular order. The primitives are defined as picking up a specific type of object. We train our segmentation model on trajectories with 2-4 skill primitives and test with 5 skill trajectories. Each instantiation of the environment has a different starting position of the agent, different starting position of the objects, and different set of objects needed to be grabbed. The image inputs used are 33 by 30 resolution color images, and the max trajectory lengths are 50. This environment serves as toy scenario for proof of concept. At the bottom-left of the image, there is an indicator which suggests which skill is being executed. Hence, an efficient approach should achieve 100% accuracy, as is the case with our method as shown in Table 1.

| Method | Train | Val | Test |
|---|---|---|---|
| Random | 20.00 | 20.00 | 20.00 |
| Random-Cls | 36.00 | 36.00 | 20.00 |
| CCNN | 65.27 | 64.63 | 60.24 |
| FCN-MIL | 91.28 | 91.88 | 91.58 |
| MIL | 90.78 | 91.08 | 91.08 |
| **Ours** | **100.00** | **100.00** | **100.00** |

Table 1: This environment serves as only a proof of concept. Since the environment is a simple 2D grid with skill primitive indicator cell at the bottom, an efficient approach should be able to achieve almost full accuracy. Our method is able to perfectly segment the data into ground truths, while other baselines can not despite easily separable skills.

### 4.2    DIAL CONTROL ENVIRONMENT: JACO ROBOTIC ARM BASED MANIPULATION

In the Dial environment, proposed in (Shiarlis et al., 2018), there is a torque-controlled JACO 6 DoF arm and a dial pad that the arm can interact with which is simulated in MuJoCo. There are naturally 10 different types of primitives available in this environment corresponding to pressing numbers zero through nine. We train our segmentation model on trajectories with two to four sub-tasks and test with five sub-task trajectories. Each instantiation of the environment has a different sequence of numbers it expects to be dialed in the correct order. The image inputs used are 112 by 112 resolution gray-scale images, and the max trajectory lengths are 100.

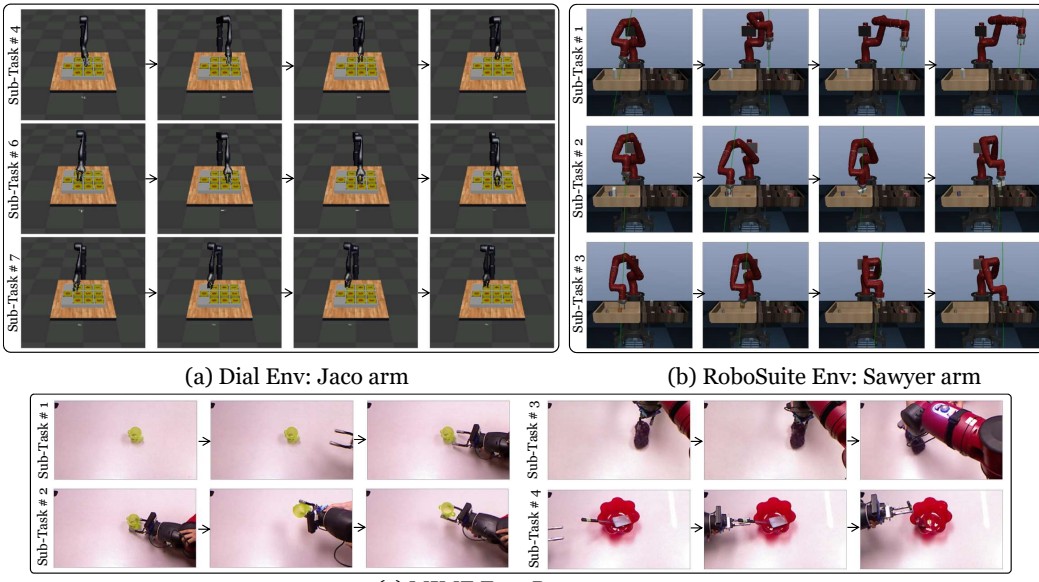

(a) Dial Env: Jaco arm        (b) RoboSuite Env: Sawyer arm

(c) MIME Env: Baxter arm

Figure 3: Figure shows qualitative visualization of the skills (sub-sampled) discovered by our approach on held-out test set. (a) Dial Env: learned primitives using Jaco arm manipulation are shown. Skill # N corresponds to the arm dialing number N on touch pad. (b) RoboSuite Env: Three predicted primitives are shown where the primitives are to pick and place an object into corresponding bin (top to bottom: cereal, milk, box) (c) MIME Env: Four discovered primitives are shown starting from top left clockwise: reach to object, wipe, stir inside object, and pour out object.

Our method performs similar to FCN-MIL on the train/val, and better on test. It significantly outperforms the other baselines (Table 2). However, we show that our segmentation are more useful for zero-shot execution performance as explained in Section 4.5. Learning perfect segmentation in the Dial environment is very challenging because there is little signal in most of the trajectory for each skill to signify exactly which digit will be pressed until the arm reaches proximity of the digit.

### 4.3 RoboSuite environment: Sawyer robotic arm based object pick and place

The RoboSuite environment (Mandlekar et al., 2018) has a Sawyer 7 DoF arm and four different objects (bread, can, cereal, and milk) that the arm can interact with simulated with MuJoCo physics engine. There are four different types of primitives available in this environment corresponding to pick and place of bread, can, cereal, and milk to correct corresponding bin. We train for trajectories with two to three skill primitives and test on trajectories with four skill primitives. Each instantiation of the environment has a different sequence and set of objects that need to be picked up and placed into their corresponding bins. The image inputs used are 128 by 128 resolution color images and the max trajectory lengths are 100.

Our method significantly outperforms all baselines in full trajectory segmentation by a significant margin. Only MIL performs above random present class on validation and test datasets. Learning perfect segmentation in the RoboSuite environment is also very challenging because there is very little signal in most of the trajectories of each subtask to signify exactly which object will be picked up until near the end of the primitive where the object has been picked up. The "pick object" portion of human demonstrations is usually much longer than the "place object" part because with the tele-operation setup the human stumbles a little bit until fully gripping the object. After the object is in the gripper, placing object in bin is a quick reach to the correct bin for the object.

### 4.4 MIME environment: Baxter Robotic Arm Based Manipulation

MIME is a robotic-demonstration dataset that contains 8260 human-robot video demonstrations of 20 different robotic tasks (Sharma et al., 2018). We defined the following primitives for a subset

| Method | Dial Jaco Manipualtion | | | RoboSuite Manipulation | | |
|---|---|---|---|---|---|---|
| | Train | Val | Test | Train | Val | Test |
| Random | 10.00 | 10.0 | 10.0 | 25.00 | 25.00 | 25.00 |
| Random-Cls | 36.00 | 36.00 | 20.00 | 33.00 | 33.00 | 25.00 |
| CCNN | 9.05 | 9.58 | 10.70 | 31.92 | 22.24 | 21.80 |
| FCN-MIL | **64.42** | **59.64** | 57.81 | 29.79 | 23.76 | 22.21 |
| MIL | 19.02 | 15.89 | 14.00 | 35.95 | 28.81 | 28.69 |
| Ours | **61.57** | **59.52** | **59.93** | **44.96** | **37.67** | **33.88** |

Table 2: We show the segmentation performance across different methods on train, validation, and test datasets for **Dial Jaco and RoboSuite Object Manipulation**. (a) Dial Jaco: we train our segmentation model on trajectories with three to four subtasks and test with five subtask trajectories. Our approach beats all baselines by significant margin except for FCN-MIL which performs similar to ours on training and validation, but is outperformed on the test set. (b) Robosuite: we train our segmentation model on trajectories with three subtasks and test with four subtask trajectories. Our approach outperforms all baselines by significant margin.

of this dataset: reach for object, pour out object, stir inside object, stack objects, place object in box, wipe with rag (6 primitives). All videos have two primitives where one is to reach for object and the other is the action to do with or on the object. There is a held out test dataset for each robotic task which we use for evaluation. The image inputs used are 120 by 320 resolution grayscale images and the max trajectory lengths are 100. Our method beats all other baselines in full trajectory segmentation by at least 1.8x on the test set (Table 3).

| Method | Test Accuracy |
|---|---|
| CCNN | 11.17 |
| FCN-MIL | 19.10 |
| MIL | 24.37 |
| Ours | **44.22** |

Table 3: Quantitative results on MIME dataset. We test our segmentation model on held out test trajectories not seen during training.

### 4.5 ZERO-SHOT RESULTS: JACO MANIPULATION

We use our segmentation model to create sub-datasets for each of our primitives to train a behavior cloned skill policy for each. We then test our skill policies on performing higher sequence length tasks not seen in training data. During the creation of the sub-datasets, we rejected all segments smaller than 5 consecutive timesteps of the same labelled primitive. We applied gaussian smoothing on the segmentation prior to extraction to filter out noisy predictions.

We demonstrate the zero-shot capability of our model and baselines on the Dial control environment in Table 4. Our model performs at least 1.25x better than all baselines. We also show that although FCN-MIL had the same segmentation accuracy as our method, after rejecting smaller than 5 timestep segments our method has a significantly higher post rejection segmentation accuracy. We speculate this is due to our model committing less to a wrong prediction than the baselines. Therefore wrong predictions are more easily rejected with our segmentation model.

## 5 RELATED WORK

Multiple Instance Learning is the sub-field that deals with learning with weak-labels (Dietterich et al., 1997), formulated mostly as max-margin. Classic formulations include MI-SVM (Andrews et al., 2002) or LSVM (Felzenszwalb et al., 2010). There have been boosting (Ali & Saenko, 2014; Zhang et al., 2005) and Noisy-OR models (Heckerman, 2013) formulations as well. This has been extensively explored in image segmentation (Pinheiro & Collobert, 2015; Pathak et al., 2015a). There is also work on learning to segment videos into primitive components that take place in each of the videos using human narration of the videos (Alayrac et al., 2018), (Naim et al., 2014), (Zhukov et al., 2019), (Richard et al., 2017). However collecting human narrations is an expensive process that cannot scale to very large datasets. In our setup, we need to label each timestep of a video with a further preferable constraint of having the labels as contiguous as possible.

Training goal conditioned policies have become very popular recently since they learn policies that are reusable for new tasks. The goals can be defined in state space (Schaul et al., 2015), (Andrychowicz

| Method | Seg Acc. without Reject | Seg Acc with Reject | Zeroshot Success Rate |
|--------|------------------------|---------------------|----------------------|
| CCNN | 9.58 | 28.08 | - |
| FCN-MIL | **59.64** | 68.58 | 38.8 |
| MIL | 15.89 | 50 | 39.6 |
| Ours | **59.52** | **79.01** | **50.4** |

Table 4: **Quantitative Zero-Shot results on Dial Jaco Manipulation**: This table shows the comparison of performance across different methods. The first two columns reports the segmentation accuracy on validation set without and with minimum of 5 time-step segment rejection. The last column shows the zero-shot subtask success rate on 5 subtask tasks. Our method has significantly better post 5 time-step segment rejection segmentation accuracy and zero-shot subtask performance.

et al., 2017) or in image space (Pathak et al., 2018), (Nair et al., 2018). These methods however suffer from not being able to reach goals that are out of the training distribution. They also tend to greedily reach goals which make them suffer for long horizon tasks. This leads to having to break the problem down to checkpoint states and this can still be expensive and cumbersome for a human to define each new task (Pathak et al., 2018). Similar way to define a new task is one-shot imitation learning where a new task is defined with one demonstration of the task; this can also be expensive if one need to learn new tasks that are cumbersome for the human to collect (Duan et al., 2017).

Hierarchical policies have been used to solve long horizon tasks by having a meta-policy or manager that takes in states at a sparser time scale and chooses which sub-policy or short term goal for worker to achieve at dense time scale (Dayan & Hinton, 1993), (Vezhnevets et al., 2017), (Sutton et al., 1999), (Bacon et al., 2017). However, the subpolicies learned in this type of options framework are not intrepretable in their specialization and therefore tend to be hard to reuse for new tasks.

Learning reusable and interpretable primitives from scratch has been very challenging. One way to tackle this is for humans to define the high level primitives we care about beforehand. We can then decompose complex tasks into subtasks / primitives and learn the primitives first (Oh et al., 2017). By learning the primitives first, an agent can then perform any new task that can be decomposed into those primitives without needing any new human demonstration data (Andreas et al., 2017). The human only needs to specify the order in which the primitives need to be done. In the presence of only the order of high level tags of what primitives were performed at the trajectory level, the optimization is non-trivial. Shiarlis et al. (2018) provide one approach using dynamic programming to efficiently compute gradients (Graves et al., 2006) through all possible trajectories of a given high level permutation of primitives performed in each trajectory. We however have presented an approach where we would not need the order in which the primitives were performed thereby making the labeling process of videos much cheaper. (Kipf et al., 2019) and (Niekum et al., 2012) do unsupervised primitive segmentation in environments using state space whereas we do weakly supervised primitive segmentation in image space in all of our environments ((Kipf et al., 2019) only use image input for their toy gridworld environment).

## 6 DISCUSSION

Obtaining primitives from existing experience and composing them to perform novel tasks presents a viable paradigm for tackling generalization to novel tasks. Due to the combinatorial nature of composition, this approach allows generalization to exponentially many scenarios with a linear number of primitives. This work provides an end-to-end trainable formulation for extracting primitives. Our results demonstrate strong segmentation accuracy, and show early evidence for zero-shot generalization via compositionality. Our primitive segmentations, obtained from demonstrations of end tasks, are more naturally extracted 'options' than those defined by an expert, which may be prone to being biased. Another alternative, besides zero-shot composition, is to treat these primitives as new atomic actions to develop a hierarchical framework for modeling long-horizon tasks. Deciding the threshold for segmenting actions into different levels of controller is one of the main bottlenecks in hierarchical control, and current approaches usually resort to using domain knowledge to solve the issue. These extracted primitives, a.k.a 'macro' actions, provide an alternative scaffolding which could bootstrap the hierarchy in a bottom-up manner. To promote these follow-up ideas, we will publicly release our code and environments.

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
