# OpenReview forum: "Weakly-Supervised Trajectory Segmentation for Learning Reusable Skills"
_ICLR.cc/2020/Conference — Reject_

### Official Review · AnonReviewer1 · 2019-10-14
**Official Blind Review #1**

**Rating:** 3

**Review:**

This paper introduces a weakly supervised learning approach for trajectory segmentation, which relies on coarse labelling about the occurrence of a skill in a demonstration to segment trajectories. This is accomplished through a recurrent model predicting skill categories for each step in a trajectory, and a trajectory level loss function that penalises the probability of seeing a given skill in a trajectory.

Overall, I like the idea of identifying skills in this manner, and think that this is an important problem to address. However, I have concerns about it's feasibility when a large number of skills are present. It seems that there are certain requirements of skill occurrences in datasets that need to be met if this approach is to be feasible. For example, consider the dataset of skill sequences:

[111 222 333]
[444 222 333]
[222 333 555]

It seems that it will never be possible to learn to identify skills 2 and 3 from this dataset. This paper would be greatly strengthened if the minimum dataset requirements to learn all skills were enumerated, or some theoretical bounds provided around when this loose labelling could possibly be successful provided.  The paper glosses over this point by suggesting that "data" makes this a non-issue, but the paper would be much stronger if these limitations were confronted and some bounds on the chances of meeting the requirements needed for learning provided.

The classification results seem extremely poor, and it is hard to assess these on the basis of accuracy alone. For example, in the Robosuit example, the test results could potentially have been obtained by simply making the same prediction throughout the test, and there is no indication that anything sensible was actually learned. Confusion matrices, or precision and recall metrics, are required to avoid this. At present it is impossible to know if these errors are due to noisy predictions, the dataset limitations described above, or simply a poor classifier.

Along these lines, qualitative results in the form of trajectory classification (say following the presentation conventions in

Bolanos, 15, https://arxiv.org/pdf/1505.01130.pdf
Ranchod, 15, https://ieeexplore.ieee.org/stamp/stamp.jsp?tp=&arnumber=7353414&tag=1

would also help to address these concerns.

A question regarding the dial jaco videos experiments, why does the classifier not predict a 5 when moving from 4 to 6? I would expect a very jumpy prediction here, but the prediction looks very smooth - is this a filtered result?

Finally, baseline experiments are limited to other weakly supervised learning segmentation approaches,  but I think comparisons with unsupervised clustering methods would also be useful.

Unfortunately, due to the lack of evidence that the proposed approach is able to learn effectively, I am inclined to reject this paper. Clarifying the bounds, and presenting stronger evidence (beyond accuracy) would make this paper stronger.


**Experience Assessment:**

I have published one or two papers in this area.

**Review Assessment: Checking Correctness Of Derivations And Theory:**

I assessed the sensibility of the derivations and theory.

**Review Assessment: Checking Correctness Of Experiments:**

I assessed the sensibility of the experiments.

**Review Assessment: Thoroughness In Paper Reading:**

I read the paper at least twice and used my best judgement in assessing the paper.

---

> ### Author Response · Authors · 2019-11-15
> **Response to AnonReviewer1**
>
> We thank the reviewer for the constructive feedback and are glad that the reviewer likes the idea of identifying skills in the way we proposed and considers our problem statement important to address. We address the concerns in detail below.
>
> R1: “It seems that there are certain requirements of skill occurrences in datasets that need to be met if this approach is to be feasible. For example, consider the dataset of skill sequences ....”
> => We would like to emphasize that our problem statement is heavily ill-posed and under-constrained as it uses a very weak supervision to extract skills from the data. For instance, in the example given by the reviewer “222 333” always occur together, so it is not incorrect to treat “222 333” as a single skill which our method will do. Hence, for an arbitrary dataset, we can’t expect such weakly supervised method to achieve full accuracy especially because there might be no signal in the dataset to separate “222” from “333”. Instead of putting any such constraints to make our dataset favourable to our approach, we just create datasets randomly and measure how far one can push the performance of learning skills with unconstrained weak supervision.
>
> R1: “The classification results seem extremely poor, and it is hard to assess these on the basis of accuracy alone.”
> => In our results section, we explain that our classification results are low because there are many ambiguous timesteps in trajectories where it isn’t clear even for a human of which skill is being performed. One example is that pick and place of object X in RoboSuite. During the pick part of object X it’s not clear which object will be picked up until the hand hovers very close to the object to be picked. The majority of timesteps of the pick and place skill in RoboSuite is during the pick up phase since it takes significant time for humans to grab the object than to drop into corresponding bin with the controls they are given in that environment.
> => This means that a pure classification accuracy is not a good measure of network’s performance. However, this stems from a fundamental fact as to what constitutes a skill, and whether ambiguous trajectory parts should be treated as one skill or another. This is still an open research question.
>
> R1: “A question regarding the dial jaco videos experiments, why does the classifier not predict a 5 when moving from 4 to 6?”
> => We use gaussian smoothing as post-processing on our segmentation results to remove noisy predictions. This is mentioned in the implementation details.
>
> R1: “I think comparisons with unsupervised clustering methods would also be useful”
> => In our preliminary experiments, we couldn’t get clustering to work well  because it is unclear in which space one should compare two images in a video  (raw pixels doesn’t work well). We will add feature-based clustering comparisons in the final version.

---

### Official Review · AnonReviewer2 · 2019-10-22
**Official Blind Review #2**

**Rating:** 3

**Review:**

This paper tackles the problem of learning to label individual timesteps of sequential data, when given labels only for the sequence as a whole. The authors take an approach derived from the multiple-instance learning (MIL) literature that involves pooling the per-timestep predictions into a sequence-level prediction and to learn the per-timestep predictions without having explicit labels. They evaluate several pooling techniques and conclude that the log-sum-exp pooling approach is superior. The learned segmentations are used to train policies for multiple control skills, and these are used to solve hierarchical control tasks where the correct skill sequence is known.

This is a good application of the MIL approach. However, I have settled on a weak reject because in my view, the novelty and results are minor.

The main point of comparison is the log-sum-exp() pooling as compared to max() and neighborhood-max() pooling. However, if I understand correctly, the log-sum-exp() approach has been used successfully in several other domains including its original domain of semantic image segmentation. So I view the novelty of the approach to be fairly low.

In addition, although the superior pooling method (which already exists in the literature) does outperform the alternatives evaluated here, the results are somewhat underwhelming, at only ~35-60% validation accuracy. How does this compare to a fully-supervised oracle method trained with per-timestep labels?

The behavioral cloning results are also fairly underwhelming, and the experiments are not very clearly described. Am I correct in my understanding that the learned skills are composed to solve a task where the correct sequence of skills is known, but is longer than the training sequences? A success rate of 50% on this task seems rather low. How does this compare, as above, to a fully-supervised oracle baseline? Why is there no success rate reported for the CCNN baseline?

I think this is a good application of weakly-supervised MIL, but I find the specific contributions to be lacking in novelty and impressiveness of results. There are several directions that I think could improve the work:
- oracle fully-supervised results, to indicate the gap between the fully- and weakly-supervised case
- more thorough baselines on the behavior task, such as Policy Sketches [1]
- perhaps the temporal aspect of the problem could be incorporated into the pooling approach more directly to produce a more novel algorithmic contribution

[1] Andreas, Jacob, Dan Klein, and Sergey Levine. "Modular multitask reinforcement learning with policy sketches." Proceedings of the 34th International Conference on Machine Learning-Volume 70. JMLR. org, 2017.

**Experience Assessment:**

I have read many papers in this area.

**Review Assessment: Checking Correctness Of Derivations And Theory:**

N/A

**Review Assessment: Checking Correctness Of Experiments:**

I assessed the sensibility of the experiments.

**Review Assessment: Thoroughness In Paper Reading:**

I read the paper at least twice and used my best judgement in assessing the paper.

---

> ### Author Response · Authors · 2019-11-15
> **Response to AnonReviewer2**
>
> We thank the reviewer for the constructive feedback and are glad that the reviewer considers our method a good application of the MIL approach. We address the concerns in detail below.
>
> R2: “more thorough baselines on the behavior task, such as Policy Sketches [1]”
> => Policy Sketches [1] uses reward signal and curriculum learning to learn the skills. This is similar to a fully supervised case since the curriculum in [1] starts with single skill sketches. Our problem statement has much weaker supervision and uses demonstrations not environment interaction with reward supervision. To our knowledge this is the first paper assuming weak supervision of only *what* skills are being performed without knowledge of permutation of skills.
>
> R2: “Why is there no success rate reported for the CCNN baseline?”
> => The CCNN collapsed to predicting only one class for jaco. So only one skill could have been trained since the rest of the skills had zero segments. This made CCNN unusable for zero-shot results.
>
> R2: “How does this compare to a fully-supervised oracle method trained with per-timestep labels?”
> => A fully supervised oracle method should achieve very-high accuracy with the same amount of data given to the weakly supervised approach. We will add this.

---

### Official Review · AnonReviewer3 · 2019-10-23
**Official Blind Review #3**

**Rating:** 1

**Review:**

The paper presents a weakly supervised method for segmentation of trajectories into sub-skills inspired by multi-instance learning (MIL) in image classification by Andrews et al. (2002). This is done via training a classifier to label each observation per time-step with the probability of skills corresponding to that observation. These predictions are then accumulated throughout the trajectory to compute the probability of the skill in that trajectory. There is only a trajectory level supervision provided which specifies which skills are present with no specification of the order in which they appear. They empirically show that their model can achieve decent skill level classification scores on multiple environments provided that there is a large variety of demonstrations provided.

In its current form, I would recommend this paper to be rejected because 1) the framing and motivation of the paper does not correspond to the results and experiments reported and hence seems misleading 2) the paper is limited in scope 3) further experiments and comparisons to relevant baselines are needed to support the claims made in the paper.

The problem that they are proposing is interesting and is of clear value, however, the paper falls short in addressing this problem. In particular, the paper is framed as a way to learn re-usable useful sub-skills that can help generalize to new situations in control. However, the method presented provides a per time-step labelling of each observation with the associated most likely sub-skill. Having the per time-step labelling of the trajectory provides no indication that the data could be useful for learning reusable skills for downstream tasks later. One very basic experiment could be to train a behaviour cloning (BC) agent on the observation-action pairs conditioned on the sub-skill (or a separate network per sub-skill) and show that the learned policies can be leveraged in solving the tasks presented. For instance, one can train a meta-controller that can switch between these learned sub-skills to successfully perform the task. Training such sub-skills from weakly supervised skill annotations has been successfully done by Shiarlis et al. (2018). It should be noted that in their setting, the annotations are ordered which simplifies the problem to some extent.

Ignoring the motivation and focusing only on the problem setting addressed which is annotation of trajectories with skill labels, the experiments seem very restrictive to me. To my understanding there are at most 4-6 primitive skills present in each environment investigated. Looking at the videos linked, it feels like there is some overfitting to the trajectories provided as for example in the case of the video with the red bowl (Reach and Stir inside Cup), the classifier is predicting 'Stirring' from the first frames where there is not much information present in the scene and 'reach to object' could also be plausible for example. It would have been nice to see the logits of predictions per classes for these examples to understand the confidence of the model particularly when some of the skills could be equally likely (e.g. first few frames).

I found the experimental setup unclear. Particularly, details regarding the task setup, how many demonstrations are needed per task/skill, architectural choices and hyper-parameters are lacking and makes experiments hard to follow and understand. I would have also liked to see more analysis regarding the segmented trajectories, particularly how consistent these predictions are through time. To my understanding there is nothing that keeps a skill annotation consistent over some period of time (e.g. the model could keep switching its prediction every time-step). This would be quite unsatisfactory if one would like to use this to actually segment sub-trajectories associated with a given skill that could be useful for training policies. They report applying Gaussian smoothing to filter out noisy predictions but there are no details provided on how this is tuned and how much this affects the quality of segmentations.

Overall, the paper seems to me very limited in its scope and experimental results. The claims made throughout the paper are not supported empirically or theoretically. There is not enough evidence for me to assess the significance of the proposed method and know whether this is indeed useful in practice.


**Experience Assessment:**

I have published one or two papers in this area.

**Review Assessment: Checking Correctness Of Derivations And Theory:**

N/A

**Review Assessment: Checking Correctness Of Experiments:**

I carefully checked the experiments.

**Review Assessment: Thoroughness In Paper Reading:**

I read the paper at least twice and used my best judgement in assessing the paper.

---

> ### Author Response · Authors · 2019-11-15
> **Response to AnonReviewer3**
>
> We thank the reviewer for the constructive feedback and are glad that the reviewer finds our problem statement proposal interesting and “of clear value”. We address the concerns in detail below.
>
> R3: “Training such sub-skills from weakly supervised skill annotations has been successfully done by Shiarlis et al. (2018)”
> => Shiarlis et al. (2018) assumes access to not only the labels but also the permutation of labels. To obtain permutation supervision, the annotator still has to go through the whole video which is not significantly less work than fully supervised labels which additionally only contain the time-step where skills change. We work with significantly less supervision of only knowing the tags. For instance, consider when we have lots of videos for a task like cooking an omelette where ordering might not be the same across different videos. If we roughly know what skills are required to do so (e.g., picking up eggs, breaking eggs, putting in a plate etc), then we can weakly-label all those videos at once without going through each of them.
> => We also argue that our problem statement of additionally not having permutation information of the sequence of skills performed is significantly more challenging. Knowing the sequence at a high level means the model needs to only learn how to shift time markers per skill to align with full segment of each skill. However, if the permutation information is not known the ordering of the time markers and the alignment are both needed.
>
> R3: “They report applying Gaussian smoothing to filter out noisy predictions but there are no details provided on how this is tuned and how much this affects the quality of segmentations”
> =? The Gaussian smoothing has a very minor effect on the segmentation accuracy of the total dataset which is why we didn’t report them. The smoothing helps connect smaller segments into larger contiguous segments which made a critical difference on what segments were rejected since we rejected segments smaller than 5 consecutive timesteps for our behavior cloning dataset aggregation of each skill. With smoothing fewer correctly labelled segments were rejected which led to higher segmentation accuracy behavior cloning data as reported in table 4.
>
> R3: “Having the per time-step labelling of the trajectory provides no indication that the data could be useful for learning reusable skills for downstream tasks later.”
> => In our setting, we assume that downstream tasks can be defined by a new sequence of skills not seen in the training data. Having a good segmentation model will lead to less noisy separation of the data into subsets where each skill is trained with independently. These trained skills are then re-usable for a new task given the sequence in which they should be used at test-time.

---

### Decision · Program_Chairs · 2019-12-19

**Decision:**

Reject

**Comment:**

The authors present a multiple instance learning-based approach that uses weak supervison (of which skills appear in any given trajectory)  to automatically segment a set of skills from demonstrations.  The reviewers had significant concerns about the significance and performance of the method, as well as the metrics used for analysis.  Most notably, neither the original paper nor the rebuttal provided a sufficient justification or fix for the lack of analysis beyond accuracy scores (as opposed to confusion matrices, precision/recall, etc), which leaves the contribution and claims of the paper unclear.  Thus, I recommend rejection at this time.